# Oral Health Status of 12-Year-Old Hani Children in the Yunnan Province of China: A Cross-Sectional Study

**DOI:** 10.3390/ijerph18105294

**Published:** 2021-05-16

**Authors:** Jieyi Chen, Ni Zhou, Biao Xu, Yuexiao Li, Shinan Zhang, Chunhung Chu

**Affiliations:** 1Guanghua School of Stomatology, Sun Yat-sen University, Guangzhou 510000, China; chenjy679@mail.sysu.edu.cn; 2School of Stomatology, Kunming Medical University, Kunming 650000, China; zhouni@kmmu.edu.cn (N.Z.); xubiao@kmmu.edu.cn (B.X.); liyuexiao@kmmu.edu.cn (Y.L.); 3Faculty of Dentistry, The University of Hong Kong, Hong Kong, China

**Keywords:** dental caries, ethnic group, minority group, child, China

## Abstract

Background: The aim of this cross-sectional study was to investigate the oral health status of Hani 12-year-old children in Yunnan, a province in Southwest China. Method: This study employed a multistage sampling method to recruit children from local primary schools. Two calibrated dentists examined the status of dental caries, gingival bleeding and dental fluorosis by adopting the diagnosis criteria recommended by the World Health Organization. A self-administrated questionnaire was distributed. The chi-square test and multivariate logistic regression were conducted. Results: This study invited 480 Hani children, and recruited 413 children (52% boys) (response rate: 86%). The dental caries prevalence was 52%, and the caries experience associated with the mean (standard deviation) decayed, missing and filled teeth (DMFT) score was 1.10 (1.46). Gingival bleeding was diagnosed in 252 children (61%), and dental fluorosis was found in two children (0.5%). The results of the regression model indicated the prevalence of dental caries were associated with sugary snacking habits (*p* = 0.002). The prevalence of gingival bleeding was related to the mother’s education level as well as the child’s monthly pocket money (*p* < 0.05). Conclusion: Dental caries and gingival bleeding were prevalent among 12-year-old Hani children in the Yunnan province in China. Dental fluorosis was uncommon.

## 1. Introduction

China is a multiethnic country with a population of more than 1.4 billion people [1]. The dominant group is Ethnic Han, which makes up 92% of the Chinese population [2]. Aside from Ethnic Han, 55 ethnic minorities are officially recognized in China [3]. These ethnic groups are culturally different, featuring unique traditions and spoken languages [4]. The majority of the minority groups reside in the western and northeastern inlands of China, particularly in underdeveloped and mountainous regions [5]. The latest census found that ethnic minorities consisted of more than 110 million people, and 18 ethnic minority groups had populations of more than one million each [2]. Hani, one of these ethnic groups, had a population of around 1.7 million [6]. 

The Hani people are members of a Lolo-speaking ethnic group who live in Southern China, Northern Laos and Vietnam. These people also go by the name of Ho in Laos. The population of Ethnic Hani is estimated to be two million worldwide, with the majority of them (85%) residing in China. According to the sixth census, nearly all of the Hani people in China (98%) live in Yunnan province, and they are distributed across the Ailao Mountains between the Mekong River and the Red River [6,7]. Protected by the rugged valleys, Hani people are still isolated, traditional and less assimilated. Hani people are famous for their remarkable terraced rice paddies and are also known for their tea culture [8]. They produce a post-fermented black tea called Pu’er, which is also regarded as one of China’s most potent medicinal teas. Pu’er is popular among the Hani people. After the establishment of the People’s Republic of China in the 1950s, the government organized linguistic experts to formulate Latin alphabets as written forms for the Hani people. This was done to expand sources of information and to develop communication in that area. The Honghe Hani Autonomous Prefecture was also enacted in 1953, but the Hani population was still underdeveloped. The Hani people’s disposable income per capita in Hani villages was reported to be 11,000 Chinese yuen (approximately US$ 1600) in 2018 [9], an amount that was lower than the national average (approximately US$ 4000) [10].

Recently, the local government launched a number of programs to improve the facilities supporting the development of the Hani community. However, related health policies are not being executed during this economic development process. Oral health is particularly important to a person’s overall health. According to the World Health Organization (WHO) recommendation, 12-year-old children have been selected as the global index age group to surveil the oral health status and international comparisons in young people [11]. By this age, all permanent teeth with the exception of the third molar are supposed to have erupted, and a reliable sample of students could easily be obtained through the school system. Thus, the information of oral health status among 12-year-old Hani children is especially important for the local government as well as dental researchers to use to frame a dental health promotion plan. Up to now, four domestic oral-health surveys have been conducted in China: surveys in the years of 1983, 1995, 2005 and 2015. The data were gathered according to provinces instead of being categorized specifically into individual ethnic groups. In the latest epidemiological survey performed in 2015, the results showed that 39% of the 12-year-old Chinese school children had experienced dental caries, 58% had periodontal bleeding and 13% had dental fluorosis [12,13,14]. Being one of the most common oral diseases, dental caries may lead to discomfort or pain, as well as affect children’s quality of life [15]. In addition, periodontal bleeding is defined as the gingiva inflammation, which can develop into periodontitis and may eventually progress to tooth loss [16]. Moreover, moderate or severe dental fluorosis is subject to attrition which may affect the contour of the tooth [17]. Nevertheless, national oral health investigations were conducted based on the geographical administration region instead of ethnicity. Therefore, limited evidence has revealed the oral health status of the 12-year-old Hani children up to today [18].

Oral health services constitute an important part of the public healthcare system. It is crucial to acquire a whole picture of the oral health conditions of Hani people to distribute oral health promotion programs. The aim of this cross-sectional study was to investigate the oral health status of the 12-year-old Hani children in Yunnan, China. The main outcomes include the statuses of dental caries, gingival bleeding and dental fluorosis. The secondary outcomes include the prevalence of dental erosion as well as tetracycline-stained teeth. The socioeconomic and behavioral factors related to oral health status were also determined.

## 2. Materials and Methods

We conducted this cross-sectional study in accordance with the Statement of Strengthening the Reporting of Observational Studies in Epidemiology (STROBE) (Appendix A) [19]. The study was conducted in 2016 after the Kunming Medical University Institutional Review Board provided ethical approval. At that time, the Education Bureau of Yunnan Province also supported the study. We distributed the invitation letters to children’s parents or their legal guardians and obtained written consents.

### 2.1. Sample Size Calculation and Sample Selection

We determined the sample size according to the formula of *n* = Z^2^ × P × (1 − P)/d^2^. In this formula, the Z statistic was based on the confidence level, P was the prevalence rate of oral diseases and the d was the precision [20]. Because no previous study was conducted on the prevalence of oral diseases among the Hani population, we estimated the prevalence rate of oral diseases (P) was 50% so that to yield a maximum sample size [20]. We set a confidence level of 95% and a confidence interval of 5% (confidence interval: 45 to 55%), and calculated the sample size to be 384 accordingly. Estimating a response rate of 80%, this study should invite at least 479 children to participate.

This study employed a multistage sampling method to recruit participants. The distribution ratio of the Hani population in the eastern and western regions was reported to be 3:2 [21]. Accordingly, this study should invite at least 287 children in the eastern region and 192 children in the western region to participate. We acquired lists of primary schools from the education bureau, and numbered primary schools sequentially. We used a list of computer-generated random numbers to choose primary schools. This study invited all 12-year-old Hani children in the selected schools to participate until the invitation number in each region had been fulfilled. The inclusion criterion was generally healthy children who had parental consent, and the exclusion criterion was children on long-term medications.

### 2.2. Questionnaire Survey

This study involved the use of a self-administrated questionnaire that was used in previous studies [22,23,24]. The questionnaire featured two parts: Part I focused on sociodemographic background information, and Part II focused on oral health–related habits. In Part I, we collected information on the child’s sex, the father’s and mother’s education levels, and the available monthly pocket money amount. In Part II, we investigated the child’s tooth brushing habits, sugary and sour food snacking habits, as well as dental attendance experience. The recruited children answered the questionnaires. A research assistant collected the questionnaires, reviewed the answers, and followed up on the missing answers.

### 2.3. Oral Examination

Two calibrated dentists performed intra-oral examinations in the primary schools. They used disposal dental mirrors, Community Periodontal Index (CPI) probes and headlights for illumination. Calibration was conducted in the same setting before the commencement of the study. Calibrated dentist investigated the statuses of dental caries, gingival bleeding, fluorosis, dental erosion and tetracycline-stained teeth. Before the oral checkups, we required children to brush their teeth.

The diagnosis criteria of dental caries, gingival bleeding and dental fluorosis were based on the WHO recommendation [11]. For dental caries, we used the decayed, missing and filled teeth (DMFT) index to measure the dental caries experience. This study recorded a tooth as a decayed tooth (DT) when dentists detected an unmistakable cavity, undermined enamel or a detectable softened floor or wall. This study regarded a tooth as a missing tooth (MT) when it was extracted because of dental caries, and this study diagnosed a tooth as a filled tooth (FT) if it was filled permanently and was without decay [11].

For assessing gingival bleeding, dentists gently inserted the tip of the CPI probe between the tooth and gingiva, then moved it along the sulcus via short upward and downward movements with a force of no more than 20 g. This study diagnosed gingival bleeding when any bleeding was presented [11]. For dental fluorosis, we adopted Dean’s index criteria for diagnosis [25]. This study recorded dental fluorosis if dentists detected any white flecks, occasional spots, paper-white areas or brown staining on the tooth surface [25].

This study adopted Basic Erosive Wear Examination (BEWE) criteria [26] for reporting dental erosion. The child’s dentition was divided into six sextants: 17–14, 13–23, 24–27, 37–34, 33–43 and 44–47. We scored each sextant by referring to the most severely affected tooth surface. This study used four scoring levels: (0) tooth surface has no loss, (1) enamel texture has initial loss, (2) less than 50% of the tooth surface has a distinct hard tissue defect and (3) over 50% of the tooth surface has a distinct hard tissue defect. We determined the patient’s risk level by adding up the total scores of all six sextants. This study adopted four risk levels: none (total BEWE score of up to 2), low (total BEWE score from 3 to 8), medium (total BEWE score from 9 to 13) and high (total BEWE score of 14 or above). This study determined tetracycline-stained teeth if the dentists detected tetracycline discoloration in daylight [27]. Ten percent of the participants were chosen randomly on the same day to evaluate the inter- and intra-examiner reliability.

### 2.4. Data Analysis

We performed data analysis by using IBM SPSS (version 25.0, IBM Corp., Armonk, NY, USA). We adopted Cohen’s κ statistics to analysed the intra- and inter-examiner reliability. We used the chi-square tests to assess the relationship between independent variables and the prevalence of oral diseases (dental caries as well as gingival bleeding). We completed multivariate logistic regression models to study the association between risk factors and the prevalence of oral diseases. Independent variables that had *p*-values of less than 0.10 in the chi-square tests were studied as potential risk factors in the multivariate logistic regression models. We removed insignificant potential risk factors from the models via backward stepwise selection until all remaining risk factors had *p*-values less than 0.05. This study set a significance level of 0.05.

## 3. Results

This study invited a total of 480 12-year-old Hani children from four primary schools in different cities and autonomous prefectures along the border to participate. All of the invited schools agreed to join (response rate: 100%), and this study recruited 413 children (52% boys) (response rate: 86%). The reasons for non-response were school absenteeism (*n* = 37, 55.6%) and a lack of parental consent (*n* = 30, 44.4%). The intra-examiner and inter-examiner reliability were excellent with the intraclass correlation above 0.90 for the assessment of dental caries, gingival bleeding, dental fluorosis, dental erosion as well as tetracycline-stained teeth. We summarized the sociodemographic information and oral health-related habits of the recruited children in Table 1.

### 3.1. Oral Health Status

This study revealed that 215 Hani children had dental caries experience with a prevalence rate of 52.1%. The mean (standard deviation [SD]) DMFT score was 1.10 (1.46), and the mean (SD) DT score was 1.02 (1.41). A large portion (93%) of the teeth that had dental caries experience were unrestored. Only a small number of children (*n* = 15) had their teeth filled permanently and without decay, and the mean FT (SD) score was 0.08 (0.44). In addition, few teeth (*n* = 2) were extracted due to caries, with the mean MT (SD) score being less than 0.01 (0.07). We found first molars to have the highest dental caries prevalence rate, which was 25.2%. Among first molars, lower first molars presented the highest dental caries prevalence rate, which was as high as 36.2%. Except for first molars, all incisors, premolars and second molars had prevalence rates lower than 5%. Lower incisors were the least affected ones with a prevalence rate of 0.1%.

For gingival bleeding, dentists detected 252 children (61.0%) with bleeding on probing. Along with the dental caries prevalence rates, we found lower first molars have the highest prevalence rate of gingival bleeding (32.8%), followed by the upper first molars (18.5%). All incisors, premolars and second molars had prevalence rates of gingival bleeding that were lower than 10%. We found the lowest prevalence rate of gingival bleeding on upper second molars at 1.4%. This study revealed that the prevalence rate of gingival bleeding was related to that of dental caries (*p* = 0.003).

This study found dental erosion in 313 children (75.8%). Two children (0.5%) had at least one sextant had a BEWE score of 3. This study found 52 children (12.5%) have at least one sextant had a BEWE score of 2. Meanwhile, 299 children (72.4%) had at least one sextant with a BEWE score of 1. The total BEWE scores of all sextants in each child ranged from 2 to 10. This study recognized a low level of risk for dental erosion among 104 children (25.1%) with total BEWE scores of up to 2. Most of the participants (*n* = 307, 74.3%) were at medium levels of risk, with the cumulative BEWE scores being between 3 and 8. Only two children (0.5%) had cumulative BEWE scores of 9 or above, and we categorized them as being at high risk for dental erosion.

This study detected two children (0.5%) with dental fluorosis, and they were from different cities/autonomous prefectures. Two other children who were diagnosed as tetracycline-stained teeth were from the same city.

### 3.2. Oral Health-Related Habits and Risk Factors of Oral Diseases

All participants returned their questionnaires, and research assistant followed up all missing answers. A considerable number of the study children (*n* = 295, 71.4%) indicated that their daily toothbrushing frequency was less than twice. Nearly half of the study children (43.3%) reported that they had daily sugary-snacking habit, and more than a quarter of the children (29.1%) replied that they had daily sour-food-snacking habit. In addition, 198 children (47.9%) reported that they had never visited a dentist.

In the chi-square tests of independent variables and dental caries prevalence, two variables had *p*-values of less than 0.10, and they were studied as potential risk factors in the multivariate logistic regression model (Table 2). In the final model, we found only sugary snacking habits were associated with the prevalence of dental caries (Table 3). Children who had daily sugary-snacking habits had a higher chance of having dental caries (odds ratio (OR) = 1.88, 95% confidence interval (CI): 1.27, 2.79, *p* = 0.002).

In the chi-square test of independent variables and gingival bleeding prevalence, three independent variables had *p*-values of less than 0.10 (Table 2). They were studied as potential risk factors in the multivariate logistic regression model. The final model revealed two risk factors were related to the gingival bleeding prevalence significantly (Table 4). Children whose mothers had post-compulsory education level and those who had monthly pocket money more than 100 RMB were more likely to have gingival bleeding (OR = 2.23, *p* = 0.012 and OR = 1.59, *p* = 0.026, respectively).

## 4. Discussion

The present study was the first investigation to determine the oral health profile among 12-year-old Hani children. We adopted the multistage sampling method for participant recruitment. Home schooling is uncommon in China. All 12-year-old children are required to enroll in primary schools and to receive compulsory education according to Article 5 of the Compulsory Education Law of the People’s Republic of China [28]. Recruiting samples via this probability sampling method is convenient and still maintains the necessary representativeness, especially for Hani people who live in remote and isolated villages. In addition, multistage sampling is relatively economical when compared with other sampling techniques [11]. However, this sampling technique may not be as precise as the simple random sampling is [11]. Despite this, the researchers recruited Hani children successfully with a sufficiently large sample size and maintained a high response rate. A sufficient sample size is essential to maintain the representativeness. Based on the sample size calculation formula, this study can yield the biggest sample size by setting the prevalence rate (P) at 50% when P within 10~90% [20]. This study estimated the prevalence rates of oral conditions among the Hani population were in the range of 10~90% after referring to the national oral health survey which also reported prevalence rates within this range [12,13,14]. Therefore, this study set the P to be 50% to maintain the representativeness. In addition, the study achieved good reliability with very good inter- examiner agreements as well as intra-examiner agreements due to the sufficient calibration between the two examiners. This study adopted a questionnaire which had been used for previous studies conducted in Yunnan and made it possible to compare the oral health-related habits in different ethnic children [22,23,24]. Moreover, because the questionnaires were filled in by the study children who were attending primary schools and a research assistant was available if the children had any questions, we supposed the study children can fully understand the questionnaire. This study revealed that the prevalence rates of dental caries, gingival bleeding as well as dental erosion were high among 12-year-old Hani children, whereas the presence of tetracycline-stained teeth and dental fluorosis were not common.

Historically, dental caries and gingival bleeding have been the most prevailing oral diseases globally. Compared with the national data, the Hani children recruited in this study presented a higher prevalence rate of dental caries (52% vs. 39%) [11]. In addition, we also found the dental caries prevalence among the Hani children is higher than that of the Lisu children in Yunnan province (52% vs 35%) [28], while it was lower that of the Naxi children who were also reside in Yunnan province (52% vs. 71%) [18]. Comparing with children of Ethnic Bonan, Dongxiang, Kirgiz, Li, Miao, Mongol, Tibetan, Tu, Tujia, Uygur, Yi, Yugur and Zhuang who resided in other province instead of Yunnan, Hani children also presented a higher prevalence rate of dental caries. When compared with children of ethnic Koreans, Hani children presented a lower dental caries prevalence rate [18]. Along with this, Hani children presented higher mean DMFT, DT and MT scores than the national average (0.86, 0.71 and >0.001, respectively) while also presenting a lower mean FT score (0.08 vs. 0.14) [12]. This revealed that the access to dental treatment for Hani children is limited. Additionally, gingival bleeding among Hani children prevails. The prevalence rate among Hani children is close to the national average (61% vs. 58%) [13], but it is lower than that of Lisu children (61% vs. 88%) [29].

This study also explored the risk factors that related to the prevalence of dental caries and gingival bleeding. The results of the regression model suggested that children with more monthly pocket money had higher chances of obtaining gingival bleeding. In ethnic minority areas, the adolescents usually live in a dormitory (the schools might be far away from their homes) with no parental supervision. Moreover, Hani people live in a subtropical area, where produce sugar. The hot climate and the readily available sugar products in their daily lives may increase the risk of dental caries [30]. It was assumed that children who had more pocket money were more likely to consume sugary snacks frequently. The lack of parental supervision might also adversely affect their oral hygiene practice. The unfavorable snacking habit, along with poor oral hygiene practice may put those children at a higher risk of dental disease. Other factors such as the availability of fluoridated toothpaste and other preventive home-based care may also related to the prevalence of oral diseases. Though fluoridated toothpaste, mouth rinse and dental floss are commercially available in China, the adoption of these oral health-care strategies among the Hani children and their potential effect on oral conditions still need to be confirmed by further studies.

The high prevalence of dental caries as well as gingival bleeding among ethnic minority children indicated the presence of a lack of awareness of oral disease prevention. Oral health inequalities may exist among the Hani children. The barrier to dental care services may account for oral health inequality among this population [31]. Prior studies had identified two major categories of the dental care barriers: (1) environmental barriers: originating from the dental care system; (2) non-environmental barriers: originating from the child and the family such as the child’s medical conditions and dental anxiety, the parents’ decision-making as well their oral health literacy [32,33]. Another cross-sectional study found the most common environmental barrier was “Dental care is too expensive,” and “Hard to find a dentist for my child nearby” was also a frequently reported barrier [33]. Hani children also encountered these barriers. Generally, the Hani children came from the low-socioeconomic-class families, which had a lower family-level disposable income per capita and a lower area-level socioeconomic class (rural and remote areas) than the national average [9,10]. Thus, economic burden was likely to be a major dental care barrier for Hani children. In addition, the parents of the Hani children had limited access to higher education. Most of their parents received only compulsory education, and just a small number of them received high school-level education or above. Thereby, those parents were likely to have poor oral health literacy, which might hinder effective decision-making. Moreover, dental treatment or preventive treatment in a modern dental clinic may be neither available nor affordable for the Hani people who inhabit the mountainous area. Nearly half of the study’s children reported that they had never visited a dentist.

In light of the above, reducing the inequalities in oral health services among the Hani children has become an essential task from a social justice standpoint. Although the national oral health survey recruited children all over China and children from different ethnic groups had the chance to be recruited, the national report did not present the data according to ethnicity. Therefore, the prevalence rates of oral conditions among different ethnic groups remained unknown, and the need for oral health services may not be well realized. In this context, the Yunnan Provincial Science and Technology Department funded this study to understand the oral health situation and the needs of ethnic minorities. The result of this study indicated oral health promotion programs targeting this ethnic group are needed. In 2016, the General Office of the State Council of China issued the Healthy China Plan for 2030. This project plans to provide oral health education, oral hygiene instruction and pit and fissure sealants to all primary school children in China [34]. The effectiveness of this national oral health promotion program for Hani children’s oral health status needs to be confirmed.

The present study also revealed that the prevalence of gingival bleeding was related to that of dental caries. This might partly be because the two oral diseases have a microbiological origin [15,16]. Evidence indicated that oral health education (including tooth brushing instruction, dietary instruction and dental visit instruction) was efficient in improving the oral health status of young children [35]. Accordingly, health education or school-based tooth brushing programs should be implemented among the target population to improve their oral health status [36].

Additionally, a large portion of the studied Hani children had dental erosion, although only two children were diagnosed with severe dental erosion, and a few with moderate dental erosion. Dental erosion is defined as acid-related loss of tooth structure. Extrinsic (such as acids in food) or intrinsic acid (endogenous acid), nutrition, saliva flow rate, abrasion, attrition and general diseases may have an influence on the presence and severity of dental erosion [37]. In this study, around one third of the children had a sour food snacking habit, which can be one of the reasons for dental erosion. Nevertheless, we still need to explore other factors such as bruxism to explain this epidemic oral condition among the Hani population by future studies. In addition, this study found two children had dental fluorosis, and they were from different study sites. Whether these were rare cases or dental fluorosis was epidemic in some villages should be further investigated. Meanwhile, this study detected two Hani children with tetracycline-stained teeth though tetracycline, an antibiotic, was not commonly used in China due to its side effects [27]. The local government should consider imposing restrictions on the prescription of this antibiotic for young children.

One limitation is that this study is unable to assess the presence of dental plaque. The reason is that to assess the color of the tooth crown for tetracycline-stained teeth and dental fluorosis, researchers instructed all study children to brush their teeth before the oral checkups. Along with this, we are unable to analyse the association between dental plaque and dental caries or gingival bleeding among the study children. Another limitation of this study is that the design of a cross-sectional study is unable to establish a cause-and-effect relationship because the risk factors and outcome are assessed simultaneously. Longitudinal studies are needed to confirm the cause-and-effect relationship between exposure and outcome. Nevertheless, this study still adds new information to the existing literature, and stakeholders can have general picture of the oral health status of 12-year-old Hani children as a result of it.

## 5. Conclusions

Dental caries and gingival bleeding were prevalent among 12-year-old Hani children in the Yunnan province of China, and most of the decayed teeth were unrestored. Dental fluorosis was not common. We recommended launching an oral health promotion scheme among Hani children.

## Figures and Tables

**Table 1 ijerph-18-05294-t001:** Sociodemographic background and oral health-related habits of the 12-year-old Hani children (N = 413).

Variables (No. of Children)	*n*	%
***Socio-demographic background***		
Sex		
Boy	216	52%
Girl	197	48%
Father’s education level		
Compulsory education level	350	85%
Post-compulsory education level	63	15%
Mother’s education level		
Compulsory education level	351	85%
Post-compulsory education level	62	15%
Monthly pocket money		
Up to 100 RMB (~US$ 15)	228	55%
More than 100 RMB (~US$ 15)	185	45%
***Oral health-related habits***		
Tooth brushing habit (daily)		
Less than twice	295	71%
Twice or more	118	29%
Sugary snacking habit		
No	234	57%
Yes	179	43%
Sour food snacking habit		
No	293	71%
Yes	120	29%

RMB: renminbi (Chinese Yuen); US$: U.S. Dollar.

**Table 2 ijerph-18-05294-t002:** Prevalence of dental caries and gingival bleeding according to sociodemographic background and oral health–related habits (N = 413).

Variables	DMFT > 0	*p*-Value	Gingival Bleeding	*p*-Value
***Socio-demographic background***				
Sex		0.381		0.293
Boy	50%		63%	
Girl	54%		58%	
Father’s education level		0.444		0.034 *
Compulsory education level	53%		59%	
Post-compulsory education level	47%		73%	
Mother’s education level		0.530		<0.010 *
Compulsory education level	53%		58%	
Post-compulsory education level	48%		76%	
Monthly pocket money		0.085		0.024 *
Up to 100 RMB (~US$ 7)	48%		57%	
More than 100 RMB (~US$ 7)	57%		67%	
***Oral health-related habits***				
Tooth brushing habit (daily)		0.901		0.503
Less than twice	52%		62%	
Twice or more	53%		58%	
Sugary snacking habit		0.002 *		0.874
No	45%		61%	
Yes	61%		62%	
Sour food snacking habit		0.750		0.246
No	53%		63%	
Yes	51%		57%	

DMFT: decayed, missing and filled teeth; RMB: renminbi (Chinese Yuen); US$: U.S. Dollar. *: *p* < 0.05.

**Table 3 ijerph-18-05294-t003:** Dental caries risk factor for the Hani children (multivariate logistic regression, N = 413).

Risk Factor	Odds Ratio (95% CI)	*p*-Value
Sugary snacking habit		
Yes	1.88 (1.27, 2.79)	0.002
No (Reference group)		

CI: confidence interval.

**Table 4 ijerph-18-05294-t004:** Gingival bleeding risk factors for the Hani children (multivariate logistic regression, N = 413).

Risk Factors	Odds Ratio (95% CI)	*p*-Value
Mother’s education level		0.012
Post-compulsory education level	2.23 (1.20, 4.15)	
Compulsory education level (reference group)		
Monthly pocket money		0.026
More than 100 RMB (~US$ 15)	1.59 (1.06, 2.38)	
Less than 100 RMB (~US$ 15) (reference group)		

CI: confidence interval; RMB: renminbi (Chinese Yuen); US$: U.S. Dollar.

## Data Availability

The datasets generated during and/or analysed during the current study are available from the corresponding author on reasonable request.

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
