# Peer review of "Oral Health Status of 12-Year-Old Hani Children in the Yunnan Province of China: A Cross-Sectional Study"

_ijerph, 2021, doi:10.3390/ijerph18105294_

Round 1

Reviewer 1 Report

I would like to thank the opportunity to evaluate the manuscript entitled "Oral health status of Hani 12-year-old children in the Yunnan province of China: a cross-sectional study", which aimed to "to investigate the oral health status of 12- year-old Hani children in Yunnan, a province in Southwest China". The manuscript brings a well-structured and clear introduction, justifying the choice of the theme. The methodology provides a detailed description of all procedures for data collection as well as the statistical analyzes conducted. Results are clearly described, and tables are self-explanatory. Discussion brings relevant data and arguments.

Although dental surveys are frequently reported in the literature, especially assessing the experience of dental caries and periodontal disease at children and adolescents, I consider such survey to be important in specific populations, especially those at greater vulnerability. These surveys support the creation of public policies and are important for improving the oral health of these populations. It is also worth emphasizing the structure of the manuscript, which is presented in a clear language and containing all the necessary information for its good understanding.

The following are two suggestions for the discussion topic:

“Additionally, a large number of the study Hani children had dental erosion, although only two children were diagnosed with severe dental erosion, and a few children had moderate erosion. It is necessary to explore the reasons for this phenomenon and to explore efficient prevention activities”: Could the authors identify possible explanations for the high rates of dental erosion in the permanent dentition of 12-years old children, such as oral or eating habits? It would be interesting to postulate some possible explanations, such as sleep bruxism.

Another limitation to be cited would be the cross-sectional design of the study.

Reviewer 2 Report

I had the opportunity of reviewing this interesting manuscript regarding oral health status in Hani children in Yunnan.

The study merits publication and is very well written.

I just have a couple of suggestions:

  1. Was the administered questionnaire validated previously? Considering that the sample was composed of people of rural areas how can you be sure that they fully understand the questionnaire?
  2. In the discussion section it could be interesting to add more comparisons with previous literature on similar populations.

Reviewer 3 Report

This study is well performed, and the manuscript is well written. However, I have few comments as follows:
The authors have well presented the clinical finding of the target population. Can the author justify how the national oral health survey did not include the ethnic groups as well as inequalities in public oral health services in the country? 
Authors could have used the national prevalence of oral diseases among 12-year-olds for the sample size calculation. Can authors discuss this? 
How about the availability of fluoridated toothpaste and other preventive home-based care among this study population? Can authors discuss more on these topics? 

Round 2

Reviewer 2 Report

The authors followed the suggestion and the manuscript is now close to be ready for acceptance.

I only have to suggest to prefer using the impersonal form changing sentences with personal pronouns (i.e. We)

Best Regards
